# Molecular Breeding for Incorporation of Submergence Tolerance and Durable Bacterial Blight Resistance into the Popular Rice Variety ‘Ranidhan’

**DOI:** 10.3390/biom13020198

**Published:** 2023-01-18

**Authors:** Shibani Mohapatra, Saumya Ranjan Barik, Prasanta K. Dash, Devidutta Lenka, Kartika Chandra Pradhan, Reshmi Raj K. R, Shakti Prakash Mohanty, Mihir Ranjan Mohanty, Ambika Sahoo, Binod Kumar Jena, Alok Kumar Panda, Debabrata Panigrahi, Sushanta Kumar Dash, Jitendriya Meher, Chitta Ranjan Sahoo, Arup Kumar Mukherjee, Lipi Das, Lambodar Behera, Sharat Kumar Pradhan

**Affiliations:** 1ICAR-National Rice Research Institute, Cuttack 753006, India; 2Environmental Science Laboratory, School of Applied Sciences, KIIT Deemed to be University, Bhubaneswar 751024, India; 3ICAR-National Institute for Plant Biotechnology, New Delhi 110012, India; 4College of Agriculture, Odisha University of Agriculture & Technology, Bhubaneswar 751003, India; 5RRTTS (OUAT), Jeypore 764001, India; 6Centre for Biotechnology, Siksha ‘O’ Anusandhan Deemed to be University, Bhubaneswar 751003, India; 7KVK, OUAT, Rayagada 765022, India; 8ICAR-Central Institute for Women in Agriculture, Bhubaneswar 751003, India; 9Indian Council of Agricultural Research, Krishi Bhavan, New Delhi 110001, India

**Keywords:** bacterial blight resistance, durable resistance, gene pyramiding, gene stacking, submergence tolerance

## Abstract

Ranidhan is a popular late-maturing rice variety of Odisha state, India. The farmers of the state suffer heavy loss in years with flash floods as the variety is sensitive to submergence. Bacterial blight (BB) disease is a major yield-limiting factor, and the variety is susceptible to the disease. BB resistance genes *Xa21*, *xa13*, and *xa5*, along with the *Sub1* QTL, for submergence stress tolerance were transferred into the variety using marker-assisted backcross breeding approach. Foreground selection using direct and closely linked markers detected the progenies carrying all four target genes in the BC_1_F_1_, BC_2_F_1_, and BC_3_F_1_ generations, and the positive progenies carrying these genes with maximum similarity to the recipient parent, Ranidhan, were backcrossed into each segregating generation. Foreground selection in the BC_1_F_1_ generation progenies detected all target genes in 11 progenies. The progeny carrying all target genes and similar to the recipient parent in terms of phenotype was backcrossed, and a total of 321 BC_2_F_1_ seeds were produced. Ten progenies carried all target genes/QTL in the BC_2_F_1_ generation. Screening of the BC_3_F_1_ progenies using markers detected 12 plants carrying the target genes. A total of 1270 BC_3_F_2_ seeds were obtained from the best BC_3_F_1_ progeny. Foreground selection in the BC_3_F_2_ progenies detected four plants carrying the target genes in the homozygous condition. The bioassay of the pyramided lines conferred very high levels of resistance to the predominant isolates of bacterial blight pathogen. These BB pyramided lines were submergence-tolerant and similar to Ranidhan in 13 agro-morphologic and grain quality traits; hence, they are likely to be adopted by farmers.

## 1. Introduction

Rice, the queen of cereals, is the livelihood of millions of the world’s population. It is a staple food crop for more than two billion people around the world. The global population is rising at a very rapid pace. India is the largest rice-growing nation in the world. By 2035, we will need about 2–3 million tons of additional milled rice/annum. The per capita agricultural land holdings are decreasing day-by-day due to the population increase. The production increase must be met using less land, labor, chemicals, and water. In addition, we have to produce more under a climate change scenario. Therefore, in the future, it will be an extremely challenging task to fulfill the gap between demand and supply [1]. Rice can be grown in various agro-climatic and demographic situations. The crop covers about 163 Mha, of which nearly 45% of the crop areas are under rainfed conditions with low productivity due to many biotic and abiotic stresses [2,3].

Among the abiotic stresses, drought, submergence, and water logging are the major constraints for higher productivity in the eastern region of the country, where >80% of the rice areas are of a rainfed ecology. The problem of frequent flash floods in this region is a common problem, which covers about 16 Mha [4]. Under submergence, rice plants face many problems for optimum growth and survival. Stress reduces the diffusion of gas, particularly inhibiting the uptake of oxygen, and restricts anaerobic respiration. Furthermore, muddy water reduces photosynthesis due to lower light visibility. For a longer submergence period of 2 weeks, rice plants face shortages of nutrients and energy. The affected plants start decaying and finally perish. Therefore, a complete crop failure may occur in the case of flash floods and submergence for more than 1 week if susceptible varieties are impacted. A major QTL *Sub 1A* is useful under such submergence stress and shows tolerance up to 14 days. Many popular high-yielding submergence-susceptible varieties have gained tolerance to this stress through incorporation of the *Sub1* QTL using marker-assisted backcross breeding [2,3,5]. High-yielding varieties carrying multiple stress tolerance genes are needed for farmers to face the frequent and unpredictable adverse effects due to climate change [5,6].

In addition to submergence stress, a main factor limiting yield is the BB disease caused by *Xanthomonas oryzae* pv. *oryzae* (*Xoo*) which is a destructive disease of lowland rice. The disease typically reduces rice yield by 30% but upto 80% depending on location, season, and variety [7]. The fodder quality is also reduced due to this disease. The leaves are covered by bacterial blight lesions, which reduce photosynthesis and, hence, drastically reduce the yield due to partial grain filling. Globally, >45 BB resistance genes have been reported from different sources [8]. Tightly linked molecular markers have been reported for most resistance genes. Meeting the projected demand for staple food grains beyond 2030 is a challenging task. The increased demand needs to be met without any adverse effects on the environment. Therefore, host plant resistance has gained enormous importance in recent times. A combination of three resistance genes (*xa5+xa13+Xa21*) was observed to be durable and very effective in controlling the disease [1,9].

Many popular rice varieties that were susceptible to many stresses have successfully been converted into multiple stress-tolerant versions with respect to submergence, drought, BB, etc. [2,3,6,10]. Ranidhan is a popular variety of Odisha but is highly susceptible to flash floods and bacterial blight disease. Therefore, the transfer of a suitable combination of BB resistance genes (*Xa21*+*xa13*+*xa5*) along with the flood tolerance QTL (*Sub1*) into this popular variety is a priority for development. This new version of Ranidhan pyramided lines featuring both abiotic (submergence) and biotic (bacterial blight) stress tolerance will be highly useful to the farmers of eastern India, where a large acreage of rainfed lowland exists. Hence, the present investigation is an attempt to introgress three BB resistance genes (*Xa21*, *xa13*, and *xa5*) and the *Sub1A* QTL into the rice variety Ranidhan through a marker-assisted breeding approach.

## 2. Materials and Methods

### 2.1. Plant Materials and Hybridization

Two donor parents, namely, Swarna-Sub1 and CR Dhan 800 (Swarna MAS), were used in this breeding program. The variety Swarna-Sub1 was used as the submergence-tolerant donor parent, while the other donor parent, CR Dhan 800, a Swarna-derived variety, was the carrier of three BB resistance genes (*xa5, xa13*, and *Xa21*). The recurrent variety used in this breeding program was Ranidhan, which is a popular variety of the late duration group but highly susceptible to BB disease and submergence stress.

Ranidhan was hybridized with CR Dhan 800 during the wet season of 2016 to produce the F_1_ seeds. The true F_1_ plant of Ranidhan × CR Dhan 800 was crossed with Swarna-Sub1, and F_1′_ seeds were produced during the dry season of 2017. In the next wet season of 2017, the true F_1′_ plant was backcrossed with Ranidhan, and BC_1_F_1_ seeds were produced. Foreground selection was performed to detect the progenies carrying all four target genes in heterozygous condition in the BC_1_F_1_, BC_2_F_1_, and BC_3_F_1_ generations, and the positive progenies carrying these genes and showing maximum similarity to the recipient parent, Ranidhan, were backcrossed into each segregating generation. The target gene carrying BC_1_F_1_ plant similar to the recipient parent was backcrossed with the recipient parent, Ranidhan, during the dry season of 2018 for BC_2_F_1_ seed production. In the wet season of 2018, BC_2_F_1_ plants were backcrossed with the recipient parent, Ranidhan, and BC_3_F_1_ seeds were produced. During the wet season of 2019, BC_3_F_2_ plants were grown in the field; homozygotes for the target traits were identified through genotyping, and BC_3_F_3_ seeds were produced. The progenies carrying all target genes and the QTL in homozygous condition were advanced for seed increase during the dry season of 2020. The pyramided lines and parents were evaluated for submergence tolerance, BB resistance, and yield, including 10 traits related to yield and grain quality during the wet seasons of 2020 and 2021 (Figure 1). Data analysis for various agro-morphologic traits was performed by following the standard procedure described in previous publications [11,12,13].

### 2.2. DNA Isolation and Polymerase Chain Reaction

The genomic DNA of the parental line and the progenies was isolated in each backcross generation following the CTAB method [14]. The reaction mixture for PCR had 30 ng of template DNA, 200 µM dNTPs, 5 pmol of each primer, 1× PCR buffer (10 mM Tris-HCL, pH 8.3, 1.5 mM MgCl_2_, 50 mM KCl, and 0.01 mg/mL gelatin), and one unit of Taq DNA polymerase. The volume of the mixture was 20 µL, and amplification of the target sequences was performed. For resolution of the amplified fragments of the PCR products, samples were loaded on 2.5% agarose gel in 1× Tris–borate ethylene diamine tetra acetic acid buffer, and the gel was run at 120 V for about 4 h. A photograph of the bands was taken using a Gel Doc System (Syn Gene, Bangalore, India). The protocols followed in this experiment for polymerase chain reaction, electrophoresis, and gel documentation followed those reported in earlier publications [15,16,17].

### 2.3. Marker Analysis

For tracking of the present target genes, seven gene-specific and linked markers available publicly were used for selection of the backcross progenies (Table 1). The markers were first validated in the donor parents for the presence of the target genes. Forward selection was performed in the progenies up to BC_3_F_2_ generations for tracking of the target genes and QTL. The marker data analysis, construction of the matrix from the binary data, dendrogram construction, and principal component analysis were performed as reported in previous publications [18,19,20].

### 2.4. Evaluation for Submergence Tolerance

The evaluation for submergence tolerance of the seven BC_3_F_4_ pyramided lines and parental lines were performed during the wet season of 2020, while that of the BC_3_F_5_ generation lines was performed during 2021. The evaluation was performed in the control screening tank of ICAR-NRRI, Cuttack. The 10 genotypes, including susceptible and resistant checks, were planted with 20 plants/row providing spacing of 20 cm × 15 cm following a complete randomized block design with three replications. The genotypes were submerged after 21 days of transplanting in the tank. Borewell water flow was maintained to submerge the plants completely in the screening tank in the shortest time. The water level of the tank was checked daily, and irrigation water was supplied to submerge the plants. The test materials were exposed to complete submergence stress for 14 days, and the water depth was maintained daily up to 1.0 m. The water from the tank was reduced to end submergence after 14 days. The draining of the tank water was performed carefully to avoid lodging of the seedlings. The regeneration percentage was calculated after 7 days of de-submergence. Recording of the observations and scoring of the genotypes were performed as described in earlier publications [1,27].

### 2.5. Bioassay for Resistance against BB Pathogen

The virulent strains of bacterial blight disease were artificially inoculated for studying the disease reactions against the pyramided and parental lines. The standard clipping method was followed for inoculation of the pathogen strains into the pyramided lines to create the epiphytotic condition. Eight highly virulent BB strains maintained at NRRI, Cuttack were inoculated into the plant seedlings at about 45 days old. The bacterial mass of bacterial blight pathogen isolates was suspended in sterile water at a concentration of approximately 10^9^ cells/mL. BB inoculation was performed in five leaves per plant for five different plants for each test genotype and from each replication during the wet seasons of 2020 and 2021. The disease lesion lengths (LLs) were measured after 15 days of inoculation. The recorded mean lesion lengths were used to classify the disease response of the parental and pyramided lines as resistant (R), moderately resistant (MR), moderately susceptible (MS), and susceptible (S) to the disease. The BB disease scoring and response of the pyramided and parental lines were grouped following the standard evaluation system (SES), IRRI [28], and the procedure was as described in earlier publications [29,30,31].

### 2.6. Evaluation of Pyramided Lines for Yield, Agro-Morphology, and Quality Traits

The 25 day old seedlings of the test genotypes along with parents were transplanted into the main field during the wet seasons of 2020 and 2021. Standard agronomic practices recommended for shallow lowland ecology including need-based plant protection measures were followed. The experiment was performed following an RBD (randomized complete block design) with three replications. Each test entry accommodated 35 plants per row, and five rows were taken per entry. A plot size of 5.25 m^2^ was established for each test genotype, and transplantation occurred with 20 cm × 15 cm spacing. Data were recorded from 10 plants of each entry and replication for yield and 13 agro-morphologic and quality traits, namely, panicle length, plant height, spikelet fertility, panicle weight, 1000-seed weight, number of seeds per panicle, grain length, grain breadth, head rice recovery (%), and amylase content (%), while the data for plot yield (t/ha) and days to 50% flowering were recorded on a whole-plot basis. Recording of observations was performed at the flowering, crop maturity, and post-harvest stages of the crop following the standard evaluation system (SES), IRRI [28]. The average length and breadth of 10 kernels were measured. The method of Tan et al. [32] was adopted for estimation of the head rice recovery. The standard method of Juliano [33] was used for estimation of the amylose content of the test genotypes. Principle component analysis (PCA) was used to estimate the Euclidean distance between genotypes and the correlation between the variables using the PAST statistical program [34]. Analysis of the variance for the traits plant height, panicle length, panicle weight, spikelet fertility, number of seeds per panicle, 1000-seed weight, grain length, grain breadth, head rice recovery (%), amylase content (%), days to 50% flowering, and plot yield (t/ha) was performed using Cropstat software 7.0 [35]. All other analyses including the figures were prepared using standard software as described in earlier publications [36,37,38].

## 3. Results

### 3.1. Molecular Marker Analysis of the Parental Lines for Submergence Tolerance and Bacterial Blight Resistance

The PCR amplification was performed using the genomic DNA of Swarna-Sub1, CR Dhan 800 (Swarna MAS), and the recipient parent, Ranidhan, with molecular markers of *Sub1A, xa5, xa13*, and *Xa21*. The parental lines showed the presence of specific resistance genes and the QTL as expected, reflected by the presence of bands for *Sub1A, xa5, xa13*, and *Xa21* (Figure 2). The donor line Swarna-Sub1 showed the presence of a specific band size of 203 bp using the marker Sub1A203, confirming the presence of the *Sub1A* allele. CR Dhan 800 showed the presence of bands of 530 bp, 490 bp, and 1000 bp, indicating the presence of BB resistance genes *xa5, xa13,* and *Xa21*, respectively (Figure 2).

### 3.2. Foreground Selection in Backcross-Derived Progenies

The tightly linked molecular markers for the BB resistance genes (*Xa21, xa13*, and *xa5*) and *Sub 1A* were used for the selection of the plants carrying the four target genes. True multiple F_1′_ plants were identified using submergence and the BB markers. A true multiple target gene carrying the F_1_ plant was hybridized with the recipient parent “Ranidhan” for generating BC_1_F_1_ seeds. A total of 396 seeds for the BC_1_F_1_ generation were generated and raised in the next season. Foreground selection was performed in progenies to select the desirable progenies carrying all target resistance genes and the QTL. Two markers for the submergence tolerance trait, Sub1A203 and Sub1BC2, were used as they showed better resolution than a single marker for confirming the presence of the target QTL. Among the 396 progenies, 142 were positive for Sub1A203 and 168 were positive for Sub1BC2 markers. It was observed in this study that 142 plants showed amplification for Sub1A203 and Sub1BC_2_. Therefore, 142 BC_1_F_1_ generation plants potentially contained the target QTL for submergence tolerance (Figure 3).

In the current investigation, 142 plants positive for the *Sub1* allele were screened for the *xa5* gene, and 68 progenies generated a positive result. A total of 25 BC_1_F_1_ derivatives tested positive for the *xa13* resistance gene out of total of 68 positive plants for the *xa5* gene. Lastly, 11 plants were found to be positive for the *Xa21* gene among these 25 plants. We screened progenies carrying desirable genes in various combinations; however, following analysis for the desirable combination of the four target genes and QTL, we identified 11 out of 396 plants. Among those 11 progenies, the best plant was selected, according to the similarity to the important phenotypic traits of Ranidhan, for the subsequent backcross with the recipient parent to generate seeds for the BC_2_F_1_ generation. A total of 321 seeds were produced for the BC_2_F_1_ generation for further screening and evaluation. All 321 seeds collected from the second backcross generation were grown. The marker-assisted backcross breeding approach was adopted, and foreground selection was performed to select the progenies containing all four target genes. The amplification of the genomic DNA of 312 progenies was performed using the markers for submergence tolerance and bacterial blight resistance. For effective screening of submergence tolerance, a positive reaction for both Sub1A203 and Sub1BC2 with another RM marker (RM 8300) was used to confirm the presence of submergence tolerance. In the genotyping for the *Sub1* QTL, 156 plants were observed to be positive for submergence tolerance by Sub1A203 marker. These 156 progenies were checked for *Xa21*, and 71 plants were found to be positive for the gene. These 71 progenies were screened for *xa13*, and 28 were found to be positive (Figure 4). Finally, these 28 plants were genotyped, and 10 were observed to be positive for the *Xa21* gene. Among these 10 plants, the plant most similar to Ranidhan was selected for a further backcross with Ranidhan.

A total of 286 BC_3_F_1_ seeds were produced for further evaluation. Of these 286 progenies, 148 were *Sub1A*-positive plants, of which 62 were positive for the *xa5* gene. From these 62 progenies, 27 plants were positive for *xa13*. Screening of these 27 plants yielded 12 plants positive for the *Xa21* gene (Figure 5; Appendix A). From these 12 plants, the plant with the maximum similarity to Ranidhan was selected and self-bred. A total of 1270 seeds from the best plant carrying all four genes and QTL were produced and planted. The genomic DNA of these 1270 BC_3_F_2_ plants was amplified. For the selection of plants with all four desirable genes, we screened all 1270 plants with Sub1A20, Sub1BC2, and RM8300 markers, and 215 plants were found to be homozygous. In the next step, we screened these 215 positive plants for *xa5*, and 47 plants were found to be homozygous. For further screening, we checked these 47 progenies for the *xa13* marker, which yielded 10 homozygous plants. A final screening was performed for the *Xa21* gene, and we obtained four plants carrying all genes for submergence tolerance and the QTL for bacterial leaf blight disease resistance (*Sub1A+xa5+xa13+Xa21*) (Figure 6). The seeds from the seven pyramided lines including the four desirable plants were self-bred and used for further evaluation.

### 3.3. Evaluation of the Pyramided and Parental Lines for Submergence Tolerance

A total of 10 genotypes were selected, comprising seven homozygous for the target gene-carrying pyramided lines and the parental lines. The pyramided lines included four lines carrying all desirable target genes/QTL (*Sub1A*, *xa5*, *xa13*, and *Xa21*), as well as three in different gene combinations. They were evaluated along with the three parents in a submergence screening tank under controlled conditions during the wet seasons of 2020 and 2021. The test genotypes were exposed to a complete submergence stress of 14 days. The regeneration ability varied from 70.6% to 94.5% in all seven pyramided lines carrying the *Sub1A* QTL, while the donor parent, Swarna-Sub1, showed a regeneration ability of 87.0% after 1 week of screening tank de-submergence (Figure 7). However, the donor parent, CR Dhan 800, and recipient parent, Ranidhan, did not show any regeneration. The *Sub1A* and BB pyramided lines were almost similar to the submergence tolerance donor parent, Swarna-Sub1, in terms of regeneration ability, while the susceptible variety, CR Dhan 800, and Ranidhan did not show regeneration and died as a result of the 14 days of submergence stress (Table 2).

### 3.4. Bioassay of the Pyramided and Parental Lines for BB Disease Resistance

Seven pyramided lines including four homozygous plants carrying three BB resistance genes, along with two donor parents (Swarna-Sub1 and CR Dhan 800) and recipient parent Ranidhan, were evaluated during the wet seasons of 2020 and 2021. The BB disease lesion length was measured after 2 weeks of *Xoo* strain inoculation. The lesion lengths were shorter in the donor parent, CR Dhan 800, and could resist the bacterial blight pathogen attack, which showed a mean lesion length of 1.93 cm (1.83–2.13 cm). However, the recipient parent, Ranidhan, exhibited a longer mean lesion length of 9.81 cm (9.50–10.08 cm). The mean lesion lengths observed in the derived lines varied from 2.14 to 2.25 cm (Table 3). The results clearly indicated that the pyramided lines with BB resistance genes exhibited higher resistance compared to the recurrent parent, Ranidhan.

### 3.5. Agro-Morphology, Grain Yield, and Quality Traits of the Developed Pyramided and Parental Lines

Seven developed pyramided lines, including four homozygous lines carrying all four target genes and QTL in the background of Ranidhan, were evaluated during the wet seasons of 2020 and 2021. The pyramided lines were included along with the donor and recipient parents in the evaluation conducted at NRRI, Cuttack. Thirteen traits comprising yield, agro-morphology, and quality attributes were recorded. According to the 2 years evaluation data, the recipient parent showed a mean grain yield of 5.89 t/ha. However, all test entries carrying the four target genes/QTLs showed a higher grain yield than the popular variety, Ranidhan, used as a recipient parent (Table 4).

The pyramided line CRSB 159-87-69-942 showed the highest grain yield of 7.17 t/ha, followed by CRSB 159-87-69-717 (6.99 t/ha), CRSB 159-87-69-546 (6.845 t/ha), and CRSB 159-87-69-285 (6.765 t/ha) (Table 4). All pyramided lines exhibited higher yields than the recurrent parent, Ranidhan. The best pyramided line, CRSB 159-87-69-942, showed a yield advantage of 21.73% and 9.22% over the recurrent and BB donor parents, respectively (Table 4; Figure 8). No penalty in yield was observed in the pyramided lines upon the incorporation of submergence tolerance and bacterial blight resistance into the recipient parent. Principal component 1 explained 49.46% of the variation, while PC2 explained 16.89%. Among the traits, the highest variation was observed for panicle length, followed by the number of grains/panicle (Figure 8). The highest-yielding pyramided line showed higher spikelet fertility than the recipient parent. The pyramided lines were also similar to the recipient parent in terms of many of the agro-morphologic traits. The comparative diagrammatic representation of the agro-morphologic traits of the pyramided lines and recipient lines for plant height, number of panicles/plant, days to 50% flowering, panicle length, number of grains/panicle, seed weight, spikelet fertility, grain length and breadth, head rice recovery, and amylose content clearly showed similarity to the popular variety (Figure 9A–H, Figure 10A–D and Figure 11; Table 4).

## 4. Discussion

Rice is mostly cultivated in eastern India as a rainfed crop, representing about 80% of the total area. This region faces adverse climatic effects such as frequent flash floods, drought stress, and high wind speed. The intensities of these adverse effects have increased in recent years and are likely to further increase in future. Ranidhan is a popular late-maturing group rice variety of Odisha state. The state is located in the eastern region of the country. This variety produces high yield in normal weather. However, the variety is highly sensitive to submergence and flash flood situations. Moreover, the variety is highly susceptible to BB disease. The Ranidhan cultivators in the state suffer heavy losses in years with flash floods. Furthermore, the loss due to BB disease is very high each year. The maturity duration of this variety is in line with the major rice areas of the state. In addition, this variety produces grain types which are preferred by the people in the state. Therefore, the development of high-yielding varieties with tolerance to submergence is important for the state. The pyramided version of the Ranidhan variety carrying the *Sub1* QTL and broad-spectrum BB resistance genes will be in demand as the recipient parent is already an adopted variety of the state.

The marker-based breeding approach is more precise and less time-consuming compared to the classical breeding approach. Incorporation of four target genes into the variety Ranidhan using conventional breeding will take more time. In addition, the cost of chemicals needed for controlling the disease will be saved by the growers. Furthermore, this will be an environment friendly approach as it eliminates the use of chemicals to control the BB pathogens. Pyramiding of the *Xa21*+*xa13*+*xa5* BB resistance gene combination will provide a broad spectrum of resistance to the attack of BB pathogens [1,7]. The development of host plant resistance through marker-assisted breeding has been published by many previous researchers in rice [2,10,28,30,31,39]. Many high-yielding BB-susceptible rice cultivars such as Samba Mahsuri [30], the stacking of Sub1 and Sub4 BB-resistant genes in Swarna [1], and the combination of the BB resistance gene + *Sub1* QTL+ yield QTL [2] have been developed through marker-assisted breeding. However, in this gene pyramiding study, we stacked three different BB resistance genes with *Sub1* QTL in the popular variety Ranidhan.

High-yielding modern varieties are not suitable for rainfed rice ecosystems located in eastern India. There is a need to introduce various abiotic stress tolerance traits, particularly submergence and drought tolerance, into the high-yielding varieties. In this breeding program, submergence tolerance was transferred into the Ranidhan variety without altering other traits. The pyramided lines showed a high regeneration ability similar to the submergence-tolerant donor parent under stress conditions (Table 2; Figure 7). A complete failure of crop is observed in flooded conditions when using the sensitive Ranidhan variety under prolonged submergence. The proposed pyramided lines can be good substitutes for Ranidhan so as to stabilize production in flood-prone rainfed areas. Many high-yielding and submergence-tolerant varieties have been developed in rice using a molecular breeding approach [1,5,6,40,41,42]. The grain filling percentage in Ranidhan is relatively low under artificial inoculation and under severe bacterial blight disease. However, the BB pyramided lines showed a higher grain filling percentage compared to the recipient parent, Ranidhan (Table 4). Similar results for higher grain chaffiness under severe bacterial blight disease were reported in earlier studies [2,43,44]. No penalty in yield was observed in the pyramided lines upon the incorporation of submergence tolerance and bacterial blight resistance into the recipient parent (Table 4). Similar results were previously reported in rice following the incorporation of multiple genes [2,3,7]. The precise transfer of deficient traits into popular rice varieties by adopting molecular breeding using molecular markers has been accomplished in earlier studies [3,8,19,29,45,46,47,48]. Ranidhan is a popular variety but is susceptible to submergence caused by flooding, as well as to BB disease. In this molecular breeding program, the developed pyramided lines were submergence-tolerant and BB-resistant. The pyramided lines carried four target genes which were transferred from two donor parents. The pyramided lines showed tolerance to submergence for 14 days, as well as expected broad-spectrum resistance to the pathogens due to the incorporation of three BB resistance genes into the popular variety. Ina previous gene stacking study on submergence tolerance and BB resistance by Das and Rao [48], the rice varieties Tapaswini and Lalat were reported. However, those two varieties are mid-maturing varieties suitable for irrigated ecology. The product developed here in using a marker-assisted breeding approach in a popular variety of the late-maturing group is best suited to eastern India. However, in other studies, marker-assisted breeding was employed for the pyramiding of target resistance genes for insect pests and diseases in rice [9,10,19,30,49,50,51,52]. Only a few reports of gene pyramiding involving the combination of multiple biotic and abiotic stress tolerance traits in rice are available.

The four elite pyramided lines carrying the submergence-tolerant QTL, CRSB 159-87-69-942, CRSB 159-87-69-717, CRSB 159-87-69-546, and CRSB 159-87-69-285 identified from the BC_3_F_2_ generation, showed better regeneration ability. The pyramided lines also showed smaller mean lesion lengths (2.14 cm to 2.25 cm) than Ranidhan (9.81 cm). Additionally, these pyramided lines were similar to the popular variety Ranidhan in terms of the main agro-morphologic features, namely, plant height, number of panicles/plant, days to 50% flowering, panicle length, number of grains/panicle, seed weight, spikelet fertility, grain length, grain breadth, head rice recovery, and amylose content (Figure 9A–H; Table 4). These similar pyramided lines were achieved following three backcrosses. This was possible due to the use of two donor varieties developed in the same variety, Swarna, through marker–assisted breeding. In addition, one of the parents of the recipient parent, Ranidhan, was also of the Swarna variety. Hence, the best pyramided line carrying the target genes for bacterial leaf blight resistance and submergence tolerance was almost identical to the recurrent parent Ranidhan and, hence, will be preferred.

## 5. Conclusions

Ranidhan is a popular variety of Odisha state but is highly sensitive to flash floods and bacterial blight disease. The developed pyramided lines carrying the *Sub1* QTL and BB resistance genes were similar to the recipient parent in terms of 10 agro-morphologic and grain quality traits. The pyramided lines will be highly suitable for the flash flood-affected areas of eastern India. The regeneration ability of the pyramided lines was on par with that of the donor parent, Swarna-Sub1. The chance of resistance breakdown by BB pathogen strains is high in the case of a single resistance gene in a breeding program. In contrast, the pyramiding of three BB resistance genes into the Ranidhan variety shows more robust resistance to BB pathogens. This resistance is both broad-spectrum and durable. The developed version of Ranidhan will be adopted easily by farmers of the region as the lines are similar to the recipient parent in terms of important morphologic and grain quality traits, and it is well suited to the rainfed rice ecology of eastern India. These pyramided lines also represent an example of combining biotic and abiotic stress resistance without any antagonistic interaction.

## Figures and Tables

**Figure 1 biomolecules-13-00198-f001:**
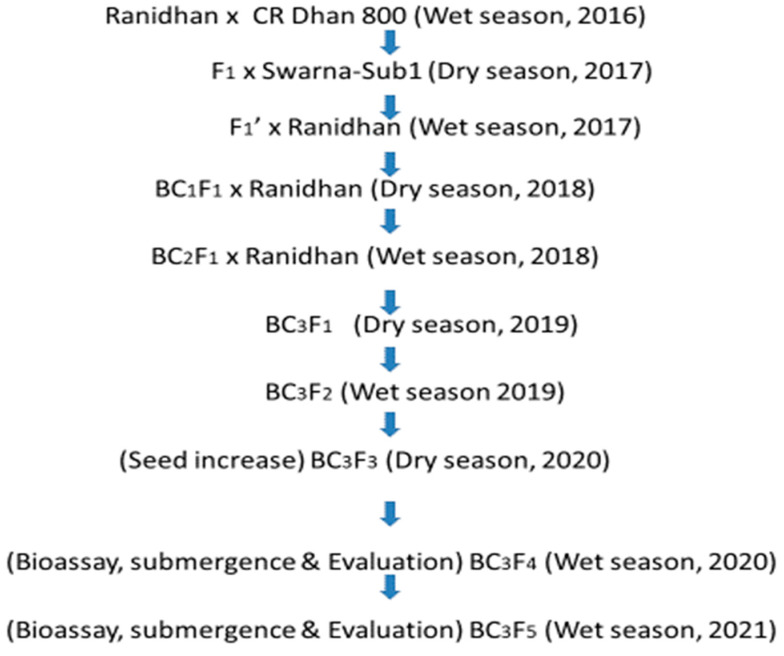
Marker-assisted breeding flowchart for transfer of submergence tolerance and bacterial blight resistance into the high-yielding popular variety “Ranidhan”.

**Figure 2 biomolecules-13-00198-f002:**
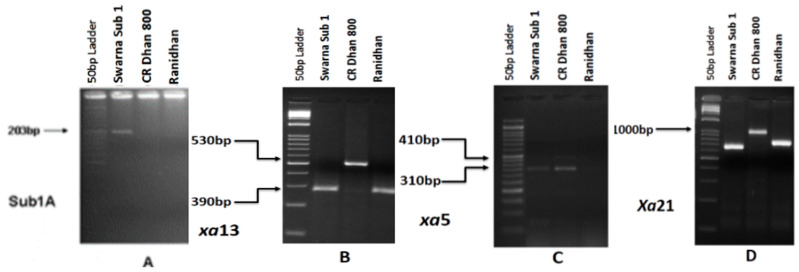
Agarose gel electrophoresis profile of parents (**A**) The presence of specific band size of 203 bp expected for *Sub1A* tolerance QTL in Swarna-Sub1 variety. Lane 1: 50 bp DNA ladder; Lane 2: Swarna-Sub1; Lane 3: CR Dhan 800; Lane 4: Ranidhan, showing the presence of *xa13* (**B**), *xa5* (**C**) and *Xa21* (**D**) with their respective sizes.

**Figure 3 biomolecules-13-00198-f003:**
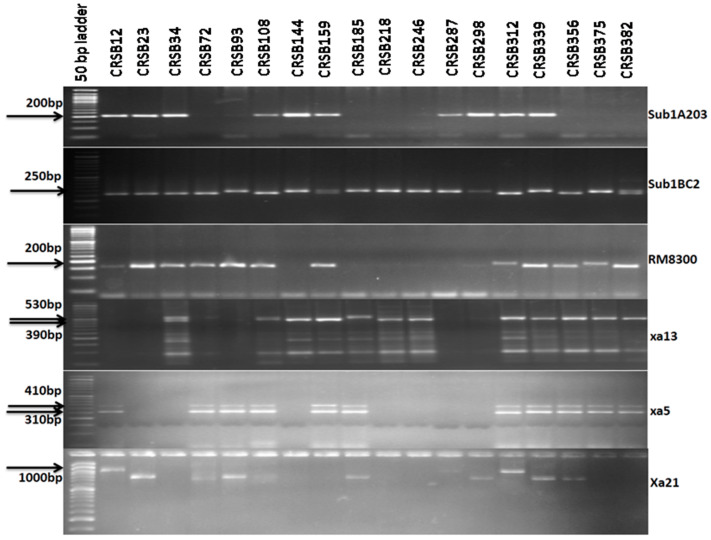
Representative electropherogram of BC_1_F_1_-derived lines of Ranidhan using submergence tolerance markers (Sub1BC2 and Sub1A203) and markers for bacterial blight resistance genes *Xa21*, *xa13*, and *xa5*. The numbers at the top indicate the BC_1_F_1_generation of Ranidhan-derived lines. A 50 bp DNA ladder was used.

**Figure 4 biomolecules-13-00198-f004:**
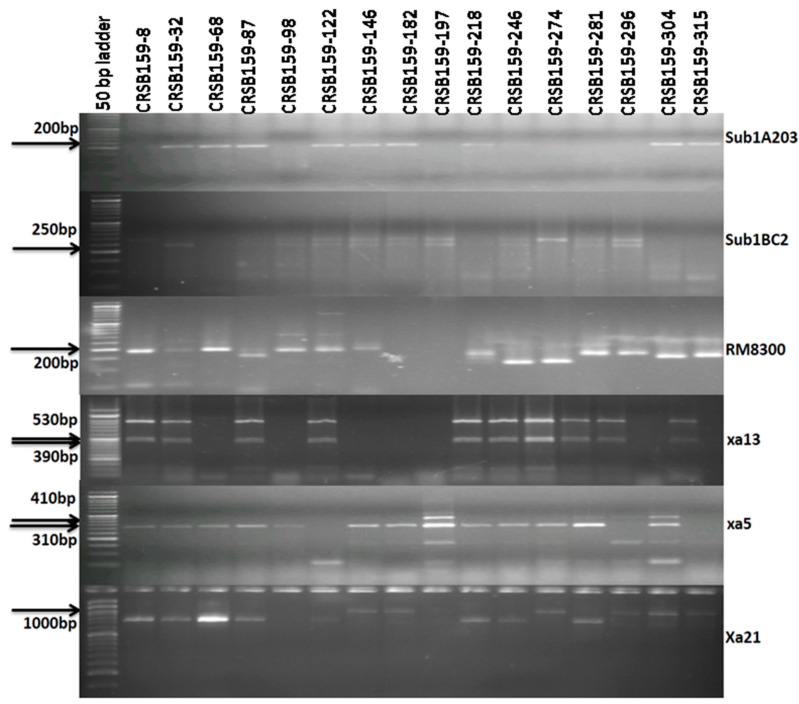
Representative electropherogram of BC_2_ F_1_-derived lines of Ranidhan using submergence tolerance markers (Sub1BC2 and Sub1A203) and markers for bacterial blight resistance genes *Xa21*, *xa13*, and *xa5*. The numbers at the top indicate the BC_1_F_1_ generation of Ranidhan-derived lines. A 50 bp DNA ladder was used.

**Figure 5 biomolecules-13-00198-f005:**
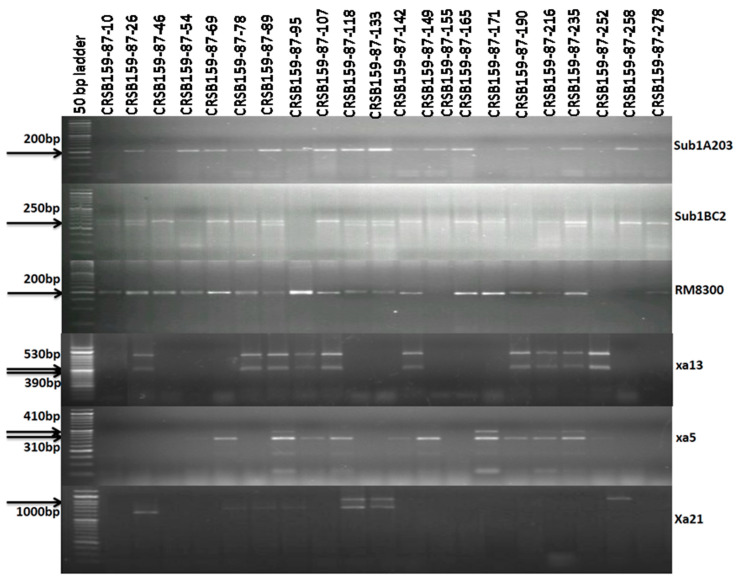
Representative electropherogram of BC_3_F_1_-derived lines of Ranidhan using submergence tolerance markers (Sub1BC2 and Sub1A203) and markers for bacterial blight resistance genes *Xa21*, *xa13*, and *xa5*. The numbers at the top indicate the BC_1_F_1_ generation of Ranidhan-derived lines. A 50 bp DNA ladder was used.

**Figure 6 biomolecules-13-00198-f006:**
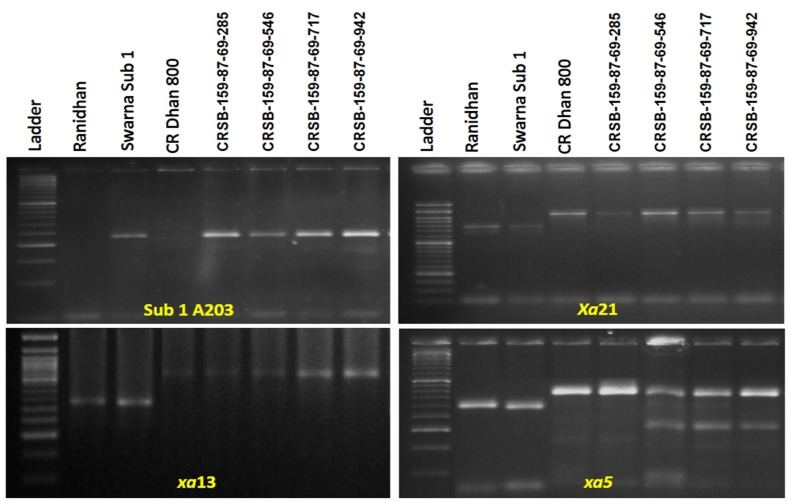
Representative electropherogram of BC_3_F_2_-derived lines of Ranidhan using submergence tolerance markers (Sub1BC2 and Sub1A203) and markers for bacterial blight resistance genes *Xa21*, *xa13*, and *xa5*. A 50 bp DNA ladder was used.

**Figure 7 biomolecules-13-00198-f007:**
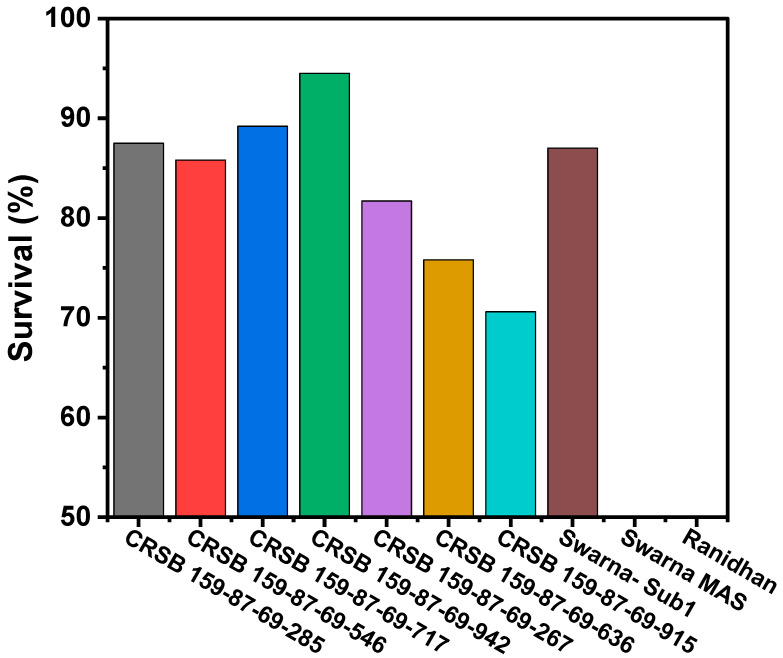
Regenerated plants (%) carrying the *Sub1* QTL in the pyramided and parental lines evaluated under the control screening conditions 1 week after 14 days of submergence stress.

**Figure 8 biomolecules-13-00198-f008:**
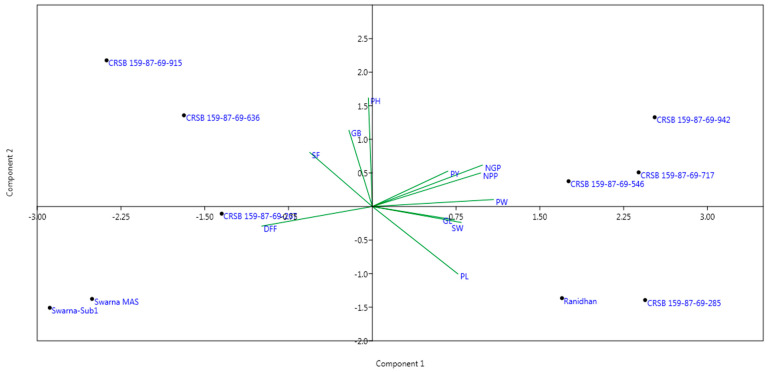
Genotype-trait biplot generated for the pyramided and parental lines using different traits: PH, plant height; NGP, number of grains/panicle; DFF, days to 50% flowering; NPP, number of panicles/plant; PL, panicle length; PW, panicle weight; SF, spikelet fertility; SW, 1000-seed weight; GL, grain length; GB, grain breadth; PY, plot yield.

**Figure 9 biomolecules-13-00198-f009:**
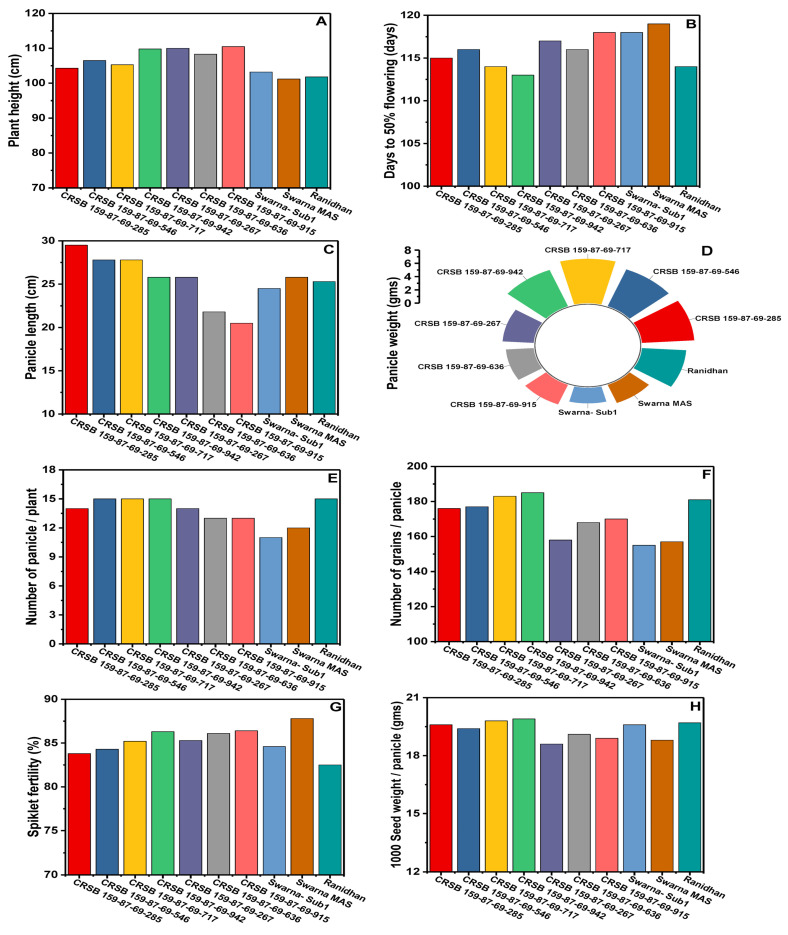
Diagrammatic representation of similarity of the pyramided and parental lines for (**A**)plant height, (**B**) days to 50% flowering, (**C**) panicle length, (**D**) panicle weight (g) (**E**) number of panicles/plant, (**F**) number of grains/panicle, (**G**) spikelet fertility, and (**H**) 1000-seed weight.

**Figure 10 biomolecules-13-00198-f010:**
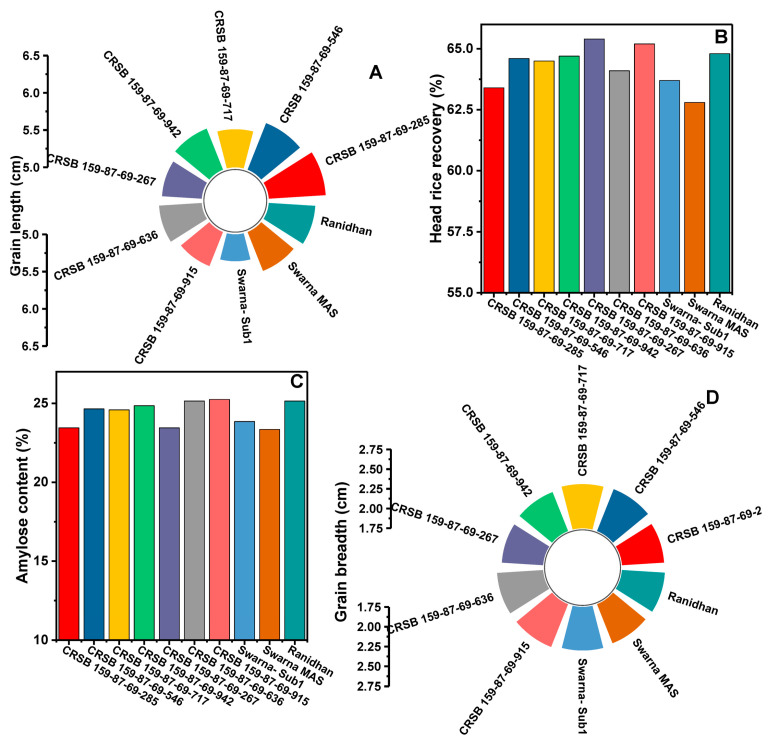
Diagrammatic representation of similarity of the pyramided and parental lines for quality traits (**A**) grain length, (**B**) head rice recovery, (**C**) amylose content, and (**D**) grain breadth.

**Figure 11 biomolecules-13-00198-f011:**
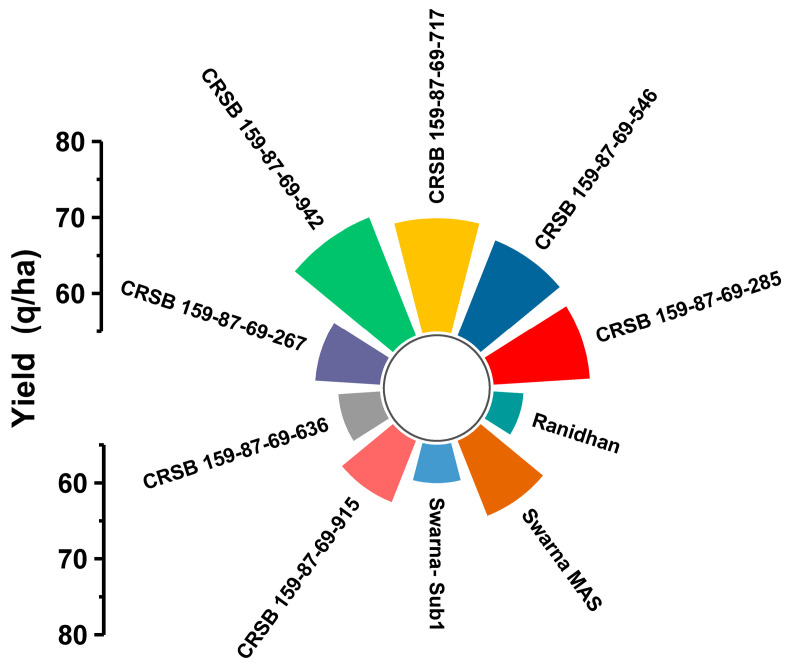
Diagrammatic representation of similarity of the pyramided and parental lines for grain yield/plot.

**Table 1 biomolecules-13-00198-t001:** Markers used for the selection of submergence tolerance QTL *Sub1* and three BB resistance genes in the backcross progenies.

Resistance Gene	Chromosome Number	Marker	Primer Sequence Used for Gene Detection		Expected Size (bp)	Marker Type	
			Forward (5′–3′)	Reverse (5′–3′)			
*xa5*	5	RM 122	GAGTCGATGTAATGTCATCAGTGC	GAAGGAGGTATCGCTTTGTTGGAC	260 bp	SSR	[21,22]
		xa5S (multiplex) xa5SR/R (multiplex)	GTCTGGAATTTGCTCGCGTTCG AGCTCGCCATTCAAGTTCTTGAG	TGGTAAAGTAGATACCTTATCAAACTGGA TGACTTGGTTCTCCAAGGCTT	160 bp	STS	[7]
*xa13*	8	Xa13 prom	TCCCAGAAAGCTACTACAGC	GCAGACTCCAGTTTGACTTC	500 bp	STS	[23,24]
*Xa21*	11	pTA248	AGACGCGGAAGGGTGGTTCCCGGA	AGACGCGGTAATCGAAGATGAAA	1000 bp	STS	[25]
*Sub1*	9	Sub1A203	CTTCTTGCTCAACGACAACG	AGGCTCCAGATGTCCATGTC	200 bp	STS	[26]
		Sub1BC2	AAAACAATGGTTCCATACGAGAC	GCCTATCAATGCGTGCTCTT	240 bp	SRS	[26]
		RM 8300	GCTAGTGCAGGGTTGACACA	CTCTGGCCGTTTCATGGTAT	205 bp	SSR	[26]

**Table 2 biomolecules-13-00198-t002:** Mean percentage survival of pyramided and parental lines submerged for 14 days, and then de-submerged and regenerated after 7 days in the BC_3_F_4_ and BC_3_F_5_ generations.

Sl. No.	Name of the Genotypes	% Survival after De-Submergence	Remarks
1	CRSB 159-87-69-285	87.5	Tolerant
2	CRSB 159-87-69-546	85.8	Tolerant
3	CRSB 159-87-69-717	89.2	Tolerant
4	CRSB 159-87-69-942	94.5	Tolerant
5	CRSB 159-87-69-267	81.7	Tolerant
6	CRSB 159-87-69-636	75.8	Tolerant
7	CRSB 159-87-69-915	70.6	Tolerant
8	CR Dhan 800	0	Susceptible
9	Swarna-Sub1	87.0	Tolerant
10	Ranidhan	0	Susceptible

**Table 3 biomolecules-13-00198-t003:** Bacterial blight disease score and reaction of the pyramided and parental lines to different *Xoo* inoculated strains during the wet seasons of 2020 and 2021.

Sl. No.	Pyramided Lines	Gene Combination	Mean Lesion Length (MLL) in cm (Mean ± Standard Error)									Disease Reaction
			*Xoo* Strains Inoculated									
			Xa17	Xa7	xa2	xb7	xc4	xd1	xa1	xa5	MLL	
1	CRSB 159-87-69-285	*xa5*+*xa13*+*Xa21*	2.03 ± 0.88	2.09 ± 0.86	2.16 ± 0.79	2.18 ± 0.77	2.31 ± 0.59	2.41 ± 0.56	2.36 ± 0.59	2.33 ± 0.62	2.23 ± 0.71	R
2	CRSB 159-87-69-546	*xa5*+*xa13*+*Xa21*	2.20 ± 0.75	2.25 ± 0.70	2.23 ± 0.63	2.13 ± 0.77	2.26 ± 0.59	2.36 ± 0.56	2.31 ± 0.59	2.28 ± 0.62	2.25 ± 0.65	R
3	CRSB 159-87-69-717	*xa5*+*xa13*+*Xa21*	2.05 ± 0.90	2.08 ± 0.84	2.13 ± 0.74	2.16 ± 0.69	2.21 ± 0.59	2.26 ± 0.56	2.23 ± 0.59	2.18 ± 0.62	2.16 ± 0.69	R
4	CRSB 159-87-69-942	*xa5*+*xa13*+*Xa21*	2.10 ± 0.85	2.05 ± 0.80	2.08 ± 0.82	2.11 ± 0.74	2.14 ± 0.59	2.21 ± 0.56	2.18 ± 0.59	2.23 ± 0.62	2.14 ± 0.70	R
5	CRSB 159-87-69-267	*xa5*	6.08 ± 0.87	6.17 ± 0.82	6.13 ± 0.72	6.26 ± 0.64	6.23 ± 0.52	6.28 ± 0.53	6.18 ± 0.52	5.93 ± 0.57	6.07 ± 0.65	MS
6	CRSB 159-87-69-636	*Xa13*	6.13 ± 0.92	6.21 ± 0.89	6.45 ± 0.75	6.45 ± 0.70	6.43 ± 0.63	6.33 ± 0.62	6.46 ± 0.69	6.38 ± 0.67	6.34 ± 0.73	MS
7	CRSB 159-87-69-915	*Xa21*	4.83 ± 1.08	4.78 ± 1.02	4.64 ± 1.11	4.84 ± 1.01	5.03 ± 0.73	5.13 ± 0.68	4.93 ± 0.73	5.10 ± 0.70	4.91 ± 0.88	MR
8	CR Dhan 800	*xa5*+*xa13*+*Xa21*	1.88 ± 0.42	1.83 ± 0.40	1.85 ± 0.30	1.97 ± 0.32	2.13 ± 0.28	2.03 ± 0.27	1.93 ± 0.28	1.84 ± 0.31	1.93 ± 0.32	R
9	Swarna-Sub1	-	10.13 ± 1.17	10.13 ± 1.12	10.11 ± 1.04	10.12 ± 0.93	10.33 ± 0.73	10.23 ± 0.72	10.13 ± 0.72	10.70 ± 0.70	10.24 ± 0.89	S
10	Ranidhan	-	9.50 ± 1.65	9.83 ± 1.62	9.80 ± 1.45	9.78 ± 1.47	10.08 ± 1.18	9.88 ± 1.18	9.86 ± 1.16	9.73 ± 1.17	9.81 ± 1.39	S

**Table 4 biomolecules-13-00198-t004:** Evaluation of pyramided lines for various yield and agro-morphologic traits in BC_3_F_4_ and BC_3_F_5_ generations.

Sl. No.	Pyramided Lines	PH (cm)	DFF (day)	PL (cm)	PW (g)	NPP	NGP	SW (g)	SF (%)	GL (cm)	GB (cm)	HRR (%)	AC (%)	PY (q/ha)
1	CRSB 159-87-69-285	104.3	115	29.5	6.15	14	176	19.6	83.8	5.76	2.28	63.4	23.45	67.65
2	CRSB 159-87-69-546	106.5	116	27.8	5.94	15	177	19.4	84.3	5.67	2.32	64.6	24.65	68.45
3	CRSB 159-87-69-717	105.3	114	27.8	6.68	15	183	19.8	85.2	5.51	2.31	64.5	24.59	69.90
4	CRSB 159-87-69-942	109.8	113	25.8	5.85	15	185	19.9	86.3	5.59	2.28	64.7	24.85	71.70
5	CRSB 159-87-69-267	110.0	117	25.8	3.65	14	158	18.6	85.3	5.53	2.27	65.4	23.45	63.50
6	CRSB 159-87-69-636	108.3	116	21.8	3.25	13	168	19.1	86.1	5.57	2.33	64.1	25.15	60.45
7	CRSB 159-87-69-915	110.5	118	20.5	3.15	13	170	18.9	86.4	5.49	2.33	65.2	25.25	63.65
8	Swarna- Sub1	103.2	118	24.5	2.34	11	155	19.6	84.6	5.36	2.30	63.7	23.85	60.03
9	Swarna MAS	101.2	119	25.8	2.92	12	157	18.8	87.8	5.56	2.28	62.8	23.35	65.60
10	Ranidhan	101.8	114	25.3	5.20	15	181	19.7	82.5	5.62	2.29	64.8	25.15	58.90
	CD_5% df_	9.57	3.23	2.47	0.75	2.18	44.57	3.07	7.84	0.25	0.07	7.614	2.816	12.13
	CV (%)	5.45	1.69	5.87	10.03	8.49	10.58	6.68	9.52	2.66	1.81	9.428	6.923	11.17

PH, plant height; DFF, days to 50% flowering; PL, panicle length; PW, panicle weight; NPP, number of panicles/plant; NGP, number of grains/panicle; SF, spikelet fertility; SW, 1000-seed weight; GL, grain length; GB, grain breadth; HRR, head rice recovery; AC, amylose content; PY, plot yield.

## Data Availability

All data generated in this study are included in the article.

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
