# Peer review of "Molecular Breeding for Incorporation of Submergence Tolerance and Durable Bacterial Blight Resistance into the Popular Rice Variety ‘Ranidhan’"

_biomolecules, 2023, doi:10.3390/biom13020198_

Round 1

Reviewer 1 Report

Authors used marker-assisted breeding approach to introgress BB resistance genes and a submergence stress QTL into Ranidhan. The work is a novel and hard-working study in production of resistant cultivar. But there is couple of suggestion to improve it:

1- Present more details (experimental design and analysis) on the evaluation of produced line during 2021-2022.

2- Give more details for PCA analysis (variability of each axis, highlighted traits in each axis…).

Author Response

Point-by-point response to the queries of Reviewer  I

Authors used marker-assisted breeding approach to introgress BB resistance genes and a submergence stress QTL into Ranidhan. The work is a novel and hard-working study in production of resistant cultivar. But there is couple of suggestion to improve it:

Query 1: Present more details (experimental design and analysis) on the evaluation of produced line during 2021-2022.

Response: Thank you. The experiment was performed in RBD (randomized complete block design) with three replications. Each test entry accommodated 35 plants per row and 5 rows were taken per entry. A plot size of 5.25 m2 was given for each test genotype and transplanted at 20 cm x 15 cm spacing. Data were recorded from 10 plants of each entry and replication for yield and 10 agro-morphologic and quality traits viz., panicle length, plant height, spikelet fertility, panicle weight, 1000-seed weight, number of seeds per panicle, grain length, grain breadth, while the data for plot yield (t/ha) and days to 50 % flowering were recorded on whole plot basis. Recording of observations were performed at flowering, crop maturity and post-harvest stages of the crop following the standard evaluation system (SES), IRRI. Average length and breadth of 10 kernels were measured. Principle component analysis (PCA) analysis was used to estimate the Euclidean distance between genotypes and the correlation between the variables using PAST statistical program. Analysis of the variance for the traits  viz., panicle length, plant height, spikelet fertility, panicle weight, 1000-seed weight, number of seeds per panicle, grain length, grain breadth, days to 50 % flowering and plot yield (t/ha) were performed using Cropstat software7.0.

We have now included details on experimental design and analysis in the revised manuscript. Kindly see the track change manuscript for the changes.

Query 2: Give more details for PCA analysis (variability of each axis, highlighted traits in each axis…).

Response: Thank you.

Principal components are new variables that are constructed as linear combinations or mixtures of the initial variables. These combinations are done in such a way that the new variables (i.e., principal components) are uncorrelated and most of the information within the initial variables is squeezed or compressed into the first components. PCA tries to put maximum possible information in the first component, then maximum remaining information in the second and so on. Principal components as new axes that provide the best angle to see and evaluate the data, so that the differences between the observations are better visible. The first principal component accounts for the largest possible variance in the data set. The second principal component is calculated in the same way, with the condition that it is uncorrelated with (i.e., perpendicular to) the first principal component and that it accounts for the next highest variance.

We have variables namely PH: Plant height; NGP: Numbers of grains/panicle; DFF: Days to 50% flowering; NPP: Numbers of panicles/plant; PL: Panicle length; PW-Panicle weight; SF: Spikelet fertility; SW: 1000 seed weight; GL: Grain Length; GB: Grain breadth & PY: Plot yield which are shown in the figure legend.

We have included the variations of PC1 and PC2 under results section. Kindly see the track change manuscript for the changes.

Reviewer 2 Report

The Manuscript entitled "Molecular breeding for incorporation of flood tolerance and durable bacterial blight resistance into popular rice variety, ‘Ranidhan’" is a nice work for pyramiding of flood tolerant genes and bacterial blight resistant genes together through MAS in an elite landrace.

The word "flood" in the title of the manuscript may be replaced with "submergence" due to various reasons. Is there any specific reason for using "durable" for bacterial resistance?

Line No. 68-69: There is probably a typographical error "my" might be "may".

In materials and methods/Plant materials section should include the agronomic practices if followed any standard or specific.

Is table No. 4 having various yield and agro- morpho traits for BC3F4 and BC3F5 generations? Has the Head Rice Recovery (HRR) been taken into account or not?

The heading in the result section "Agro-morphologic, grain yield and quality traits of the developed pyramided and parental lines" while in method and materials only "Evaluation of pyramided lines for yield and agro-morphological traits" have been described.

The rice has enormous diversity due to its diverse preference and acceptability of the people for various preference and quality related traits, as the toughest challenge for breeders during MAS is to retain the original qualities of the landrace or variety. Is there any study that has been conducted during the development and selection process specifically for preference and quality traits? If any should be included in the manuscript.

Figures of each generation of all the parents and progenies should be supplemented. It will enhance the readability of the manuscript.

The pictures of screening for "Evaluation for submergence tolerance" and "Bioassay for resistance against BB pathogen" can be supplemented.

Is there any physiological or biochemical assays have been performed for molecular characterization of the pyramided traits? If any should be included in the manuscript.

At various places the words are joined together due to no spacing, kindly check it thoroughly.

The reference are not according to the standards of the journal.

Author Response

Point-by point response to the queries of Reviewer 2

The Manuscript entitled "Molecular breeding for incorporation of flood tolerance and durable bacterial blight resistance into popular rice variety, ‘Ranidhan’" is a nice work for pyramiding of flood tolerant genes and bacterial blight resistant genes together through MAS in an elite landrace.

Query 1:The word "flood" in the title of the manuscript may be replaced with "submergence" due to various reasons. Is there any specific reason for using "durable" for bacterial resistance?

Response: Thank you. We are changing the term flood to submergence. For the durable resistance, we have pyramided 3 bacterial blight resistance genes carrying one dominant gene +two recessive genes. If pathogen mutate and break down one gene then other two resistance genes work. Therefore, we have used the term durable resistance.

Query 2: Line No. 68-69: There is probably a typographical error "my" might be "may".

Response: thank you. We have corrected it.

Query 3: In materials and methods/Plant materials section should include the agronomic practices if followed any standard or specific.

Response: We have followed the standard agronomic practices recommended for shallow lowland ecology. We have included this sentence under material and method section in the revised manuscript.

Query 4: Is table No. 4 having various yield and agro- morpho traits for BC3F4 and BC3F5 generations? Has the Head Rice Recovery (HRR) been taken into account or not?

Response: The 7 developed   pyramided lines were evaluated in two wet seasons i.e., wet season, 2020 using the BC3F4 and wet season, 2021 using the BC3F5 pyramided lines for various yield and agro-morpho traits. We have now included the head rice recovery data of the pyramided lines.

Query 5: The heading in the result section "Agro-morphologic, grain yield and quality traits of the developed pyramided and parental lines" while in method and materials only "Evaluation of pyramided lines for yield and agro-morphological traits" have been described.

Response: Thank you. We have added the term, quality also under material method section in the revised manuscript.

Query 6: The rice has enormous diversity due to its diverse preference and acceptability of the people for various preference and quality related traits, as the toughest challenge for breeders during MAS is to retain the original qualities of the landrace or variety. Is there any study that has been conducted during the development and selection process specifically for preference and quality traits? If any should be included in the manuscript.

Response: We have now included the head rice recovery % and amylose content of the pyramided lines. Now, a total of 4 quality traits namely Grain length, rain breadth, head rice recovery % and  amylose content are included in the revised manuscript.

Query 7: Figures of each generation of all the parents and progenies should be supplemented. It will enhance the readability of the manuscript.

Response: Gel images for molecular screening of traits for each generation along with the screening, bacterial blight inoculation and hybridization photos are included as Supplementary figures.

Query 8: The pictures of screening for "Evaluation for submergence tolerance" and "Bioassay for resistance against BB pathogen" can be supplemented.

Response: we have added Supplementary figures for Evaluation for submergence tolerance" and "Bioassay for resistance against BB pathogen.

Query 9: Is there any physiological or biochemical assays have been performed for molecular characterization of the pyramided traits? If any should be included in the manuscript.

Response: Biochemical assay for amylase content of the pyramided and parental lines are supplied in the Table 4.

Query 10: At various places the words are joined together due to no spacing, kindly check it thoroughly.

Response: Thanks. We have checked the specing issue.

Query 11: The references are not according to the standards of the journal.

Response: Now high standard 10 more references are added in the reference section.

Round 2

Reviewer 2 Report

The reference style of the manuscript is not in required format.